# A Study on the Effects of Gallium Droplet Consumption and Post Growth Annealing on Te-Doped GaAs Nanowire Properties Grown by Self-Catalyzed Molecular Beam Epitaxy

Shisir Devkota [1], Mehul Parakh [1], Priyanka Ramaswamy [2], Hirandeep Kuchoor [1], Aubrey Penn [3,4], Lewis Reynolds [4] and Shanthi Iyer [1,*]

[1] Nanoengineering, Joint School of Nanoscience and Nanoengineering, North Carolina A&T State University, Greensboro, NC 27401, USA; sdevkota@aggies.ncat.edu (S.D.); mkparakh@aggies.ncat.edu (M.P.); hkuchoor@aggies.ncat.edu (H.K.)

[2] Department of Electrical and Computer Engineering, North Carolina A&T State University, Greensboro, NC 27411, USA; pramaswamy@aggies.ncat.edu

[3] MIT.nano, Massachusetts Institute of Technology, Cambridge, MA 02139, USA; anpenn@ncsu.edu

[4] Analytical Instrumentation Facility, North Carolina State University, Raleigh, NC 27695, USA; clreynol@ncsu.edu

[*] Correspondence: iyer@ncat.edu

**Abstract:** In this work, the effects of arsenic (As) flux used during gallium (Ga) seed droplet consumption and the post-growth annealing on the optical, electrical, and microstructural properties of self-catalyzed molecular beam epitaxially grown tellurium (Te)-doped GaAs nanowires (NWs) have been investigated using a variety of characterization techniques. NWs using the same amount of As flux for growth of the seed droplet consumption demonstrated reduced density of stacking faults at the NW tip, with four-fold enhancement in the 4K photoluminescence (PL) intensity and increased single nanowire photocurrent over their higher As flux droplet consumption counterparts. Post-growth annealed NWs exhibited an additional low-energy PL peak at 1.31 eV that significantly reduced the overall PL intensity. The origin of this lower energy peak is assigned to a photocarrier transition from the conduction band to the annealing assisted Te-induced complex acceptor state ($Te_{As}V_{Ga}^{-}$). In addition, post-growth annealing demonstrated a detrimental impact on the electrical properties of the Te-doped GaAs NWs, as revealed by suppressed single nanowire (SNW) and ensemble NW photocurrent, with a consequent enhanced low-frequency noise level compared to as-grown doped NWs. This work demonstrates that each parameter in the growth space must be carefully examined to successfully grow self-catalyzed Te-doped NWs of high quality and is not a simple extension of the growth of corresponding intrinsic NWs.

**Keywords:** nanowires; molecular beam epitaxy (MBE); Te-doping; droplet conumption; in-situ annealing; acceptor states; optoelectronic properties

## 1. Introduction

Semiconductor nanowires (NWs) are attractive building blocks for nanostructured optoelectronic device applications, as their one-dimensional structure relaxes lattice mismatch constraints [1], and realization of a range of architectures that are not feasible in planar configurations [2,3]. These features enable integration with a wide range of substrates, high phase purity and quantum confinement effects [4,5]. In addition, the combination of sub-wavelength optical phenomenon, coupled with other optical trapping features [6,7], have contributed immensely to wide-ranging optoelectronic nano-device applications with improved performance, namely solar cells [8,9], photodetectors [10,11], sensors [12,13], and high mobility field-effect devices [14,15].

The NW growth of III-V material systems and their doping characteristics have been the subject of extensive investigation, due to their direct bandgap, enhanced optical absorption, and high carrier mobility [16–19]. Growth of high-density vertical NWs by the self-catalyzed vapor-liquid-solid (VLS) mechanism, that has been widely accepted in recent years, calls for extensive growth recipe optimization, due to many interdependent parameters influencing growth [20]. The advantage of self-catalyzed growth of NWs over foreign seed particle catalyzed growth has been well documented in the literature [21,22]. Though there has been extensive study on the effects of NW growth parameters, namely substrate temperature, beam equivalent pressure (BEP) ratios, Ga opening duration [20], dopant cell temperature [23,24] on the quality of self-catalyzed growth and NW density, dopant incorporation in NWs, the effect of droplet consumption on the NW optoelectronic properties has not been addressed sufficiently. A study on the effect of change of As flux during the catalyst droplet consumption approach revealed that the NW microstructural and morphological properties were strongly dictated by the As flux [25]. Similarly, utilization of higher As flux during the droplet consumption resulted in the NW crystallographic phase transition from zinc-blende to wurtzite [26] Post-growth in-situ annealing of the NWs is another process that has often been used to improve the quality of the NWs, particularly in antimonides [26] and antimonide-nitride material systems [11,27]. Availability of the additional thermal energy during in-situ annealing for intermixing of the constituent elements is believed to improve the optoelectronic properties [28] However, there has been little study of annealing on doped NWs, and it has largely been assumed that they follow the same trend as intrinsic NWs.

In this work, we have investigated the influence of Ga droplet consumption and post-growth in-situ annealing effects on the optical, electrical, and microstructural properties of Te-doped GaAs NWs grown by a self-catalyzed vapor liquid solid (VLS) growth mechanism using molecular beam epitaxy. The choice of Te over other commonly used dopants, namely Si, Sn, S, and Se, stems from its higher solubility in the Ga droplet leading to higher incorporation efficiency, lower diffusion coefficient, negligible cell crosstalk and memory effects [29–31]. Also, Te from a GaTe captive cell has been shown to produce reliable doping in the NWs. In this work, a systematic study on the Te doped GaAs has revealed that the amount of As flux used for consumption before the termination of the NW core growth dictates the density of stacking faults at the NW tip, thereby affecting the optical and electrical characteristics. In addition, in-situ post-growth annealing was not found to be suitable for doped NWs as it promotes the formation of Te-induced defect complexes that adversely impact optoelectronic properties.

## 2. Result and Discussions

### 2.1. Droplet Consumption

Seed droplet consumption is a requisite for the termination of the VLS growth mechanism after the desired core length is achieved before the passivation shell growth around the NW core. The NW tip of 20 nm length grown during the consumption period is a critical segment of the NW core that significantly impacts optical absorption and electrical conduction in the single and ensemble NW configurations. Generally, the seed droplet is consumed under As-rich conditions by terminating the Ga flux and, at the same time, keeping the supply of remaining growth constituent invariant. Three different As fluxes for droplet consumption were investigated for these self-catalyzed nanowires: retaining the same As flux for consumption as during growth, and half and double the growth flux. On the lower end, for half the As growth flux, droplet consumption was ineffective even for a long consumption duration of 8 min. In contrast, the droplet was consumed effectively employing the other two As fluxes, and hence only the effects of these two were the subjects of detailed study. The NWs consumed using the same As flux are identified as sample A, while those consumed with double the As growth flux are identified as sample B.

The 4K PL spectrum, depicted in Figure 1a, revealed that optical emission from the NWs using the same As flux for droplet consumption as for growth resulted in three-fold

enhancement in the PL intensity with full width at half maximum (FWHM) reduction from 96 meV to 46 meV compared to its higher As flux counterpart (sample B). Similarly, the single nanowire (SNW) photocurrent of sample A exhibited better photoconduction in both forward and reverse bias, as evidenced by the significantly higher photocurrent (Figure 1b).

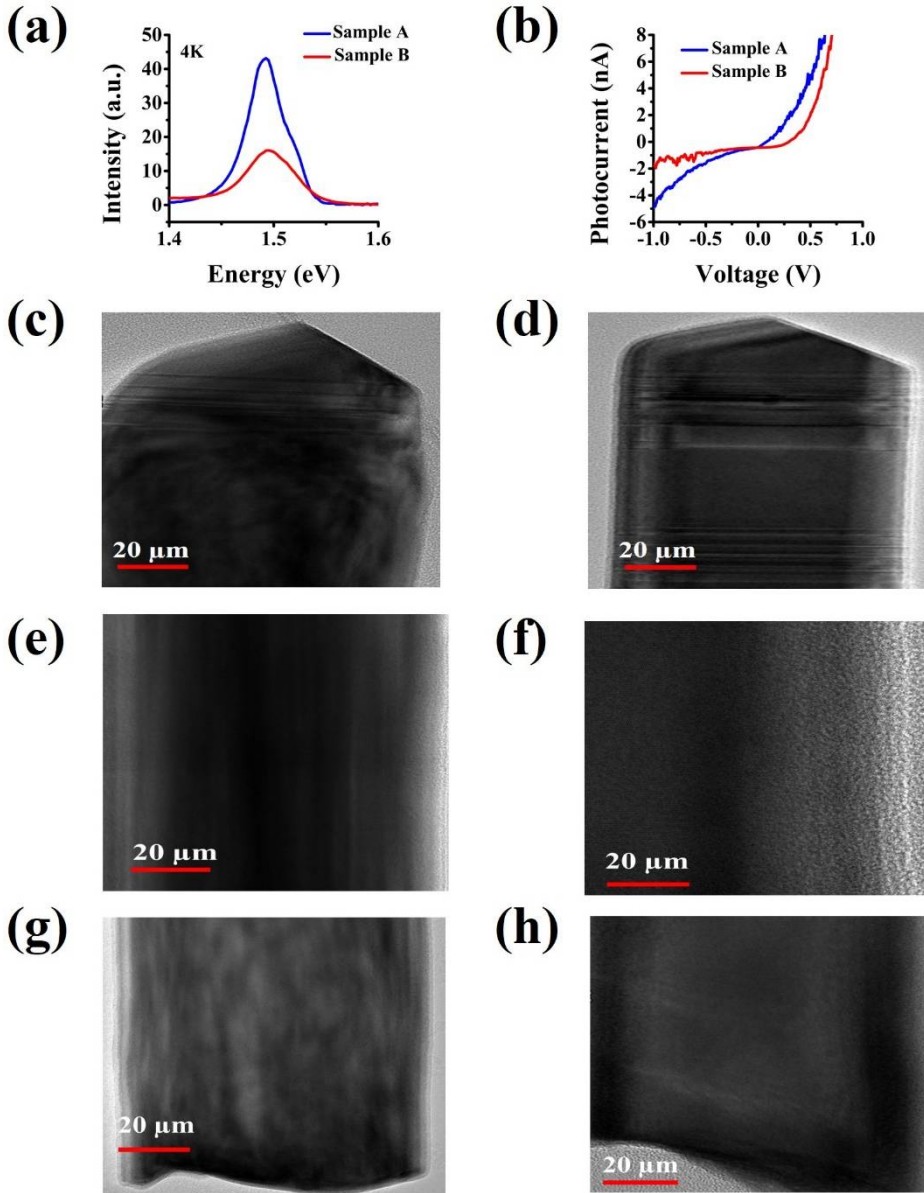

**Figure 1.** (**a**) Superimposed 4K PL spectra, (**b**) superimposed SNW I-V characteristics under illumination, (**c**,**e**,**g**) and (**d**,**f**,**h**) HRTEM images of NW top, middle, and bottom segment of Te-doped GaAs NWs with two different droplet consumption approaches.

HRTEM images of these two samples revealed a higher density of stacking faults at the tip in sample B, as represented by horizontal contrast stripes in Figure 1d, compared to sample A, in Figure 1c. However, the NW bottom and middle segments were devoid of any planar defects in both samples (Figure 1e–h); this suggests that these stacking fault defects at the tips were introduced during droplet consumption. These defects very likely are responsible for creating non-radiative recombination centers in the bandgap, suppressing the overall PL emission in sample B. In addition, potential fluctuations caused by these defects contribute to the observed increased PL FWHM (see Figure 1a).

Therefore, utilization of the same flux as the growth during droplet consumption is an effective VLS NW growth termination approach with suppressed defect density for these self-catalyzed GaAs nanowires.

### 2.2. Annealing Effects

Next, we investigated the effects of post-growth in-situ annealing on the structural, optical, and electrical properties of the Te-doped NWs using FESEM, 4K PL, and I-V characteristics. Our previous work on self-catalyzed GaAsSb NWs showed a significant influence of annealing on the optoelectronic properties, which were attributed to atomic intermixing caused by additional thermal energy [28,32].

The average length and diameter of n-GaAs NWs grown were 1.75 μm and 153 nm, respectively, with no effects of annealing on axial and radial growth, NW density or morphology, as displayed in SEM images of as-grown (Figure 2a) and annealed NWs (Figure 2b).

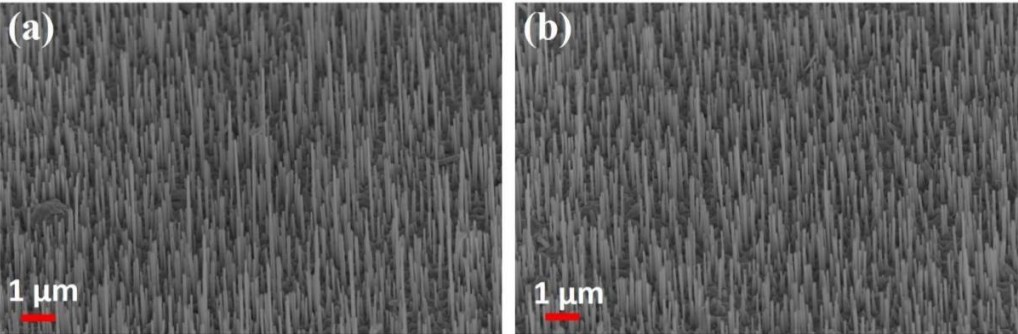

**Figure 2.** SEM images of (**a**) as-grown and (**b**) annealed n-GaAs nanowires with the As flux during droplet consumption the same as growth.

Figure 3a,b depict the 4K PL emission comparison of the annealed intrinsic (I) and annealed Te-doped (samples D) NW samples, grown under identical conditions to sample B. The value of n-type carrier concentration in the Te-doped GaAs NWs was calculated to be ~$5 \times 10^{19}$ cm$^{-3}$ based on electrical characterization and various surface measurement techniques [27,33]. The PL emission of annealed sample D exhibited an additional lower energy peak at 1.35 eV in addition to the intrinsic bandgap peak at 1.49 eV, and PL emission was also significantly enhanced (~4×) for the as-grown doped sample. The presence of a lower energy peak 0.14 eV below the intrinsic GaAs peak in sample D is assigned to a transition of photoexcited electrons from the conduction band to an acceptor level complex ($Te_{As}V_{Ga}^{-}$) formed in Te-doped GaAs [34] materials, as depicted in the band diagram (Figure 3c). The origin of this complex is speculated to arise from the interaction of a Ga vacancy with Te atoms occupying an As lattice site ($Te_{As}$) with post-growth annealing promoting the interaction, consistent with the observation in annealed Te-doped GaAs thin films [35]. An absence of this lower energy peak in intrinsic GaAs NWs (Figure 3a) and 4-fold 4K PL emission intensity reduction in annealed doped NWs relative to doped as-grown NWs (inset Figure 3b) suggest annealing-induced defect formation.

The effect of post-growth annealing on the electrical properties of the doped NWs is captured in the room temperature (RT) I-V characteristics of SNW (Figure 4a) and ensemble Te-doped GaAs NWs (Figure 4b,c). A reduction in photocurrent of the doped SNW was observed with annealing. In the ensemble configuration, as-grown Te-doped NWs displayed rectifying dark I-V characteristics with positive photo-response. In contrast, the annealed ones exhibited leaky dark I-V characteristics with negative photo-response. The low-frequency noise (LFN) spectra in Figure 4d demonstrated a higher (by six orders of magnitude) noise level in annealed NWs than as-grown NWs. All these changes suggest the creation of annealing induced Te-complex defect states, very likely being slow electron traps, that adversely impact the overall dark current [35]. Photogenerated holes have a propensity for recombination with thermal electrons resulting in a reduced free carrier

concentration [36] that leads to negative photoconductivity with trap-induced potential fluctuations elevating the noise floor.

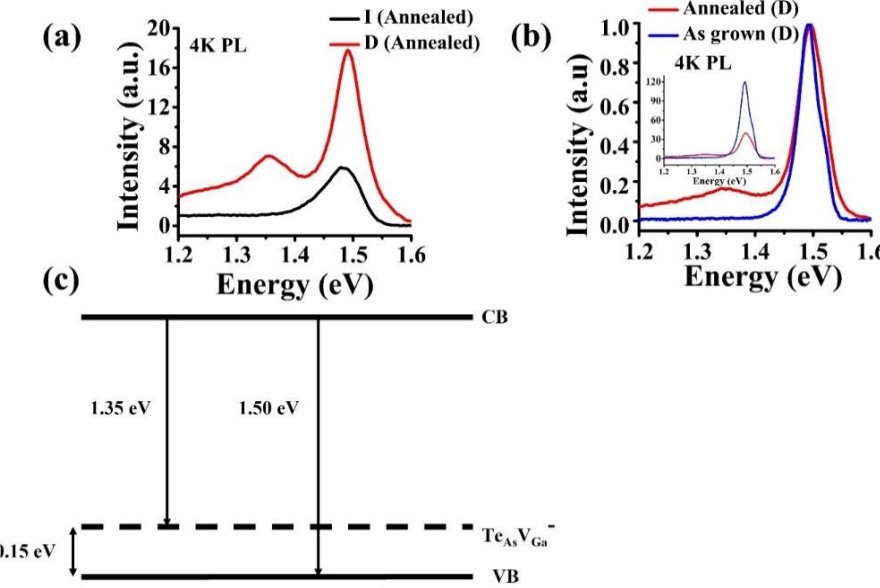

**Figure 3.** (**a**) 4K PL spectra of annealed intrinsic and Te-doped GaAs using a GaTe source temperature of 550 °C, (**b**) comparison of unnormalized and normalized 4K PL of as-grown and annealed doped NWs, respectively, and (**c**) band diagram of n-GaAs NWs.

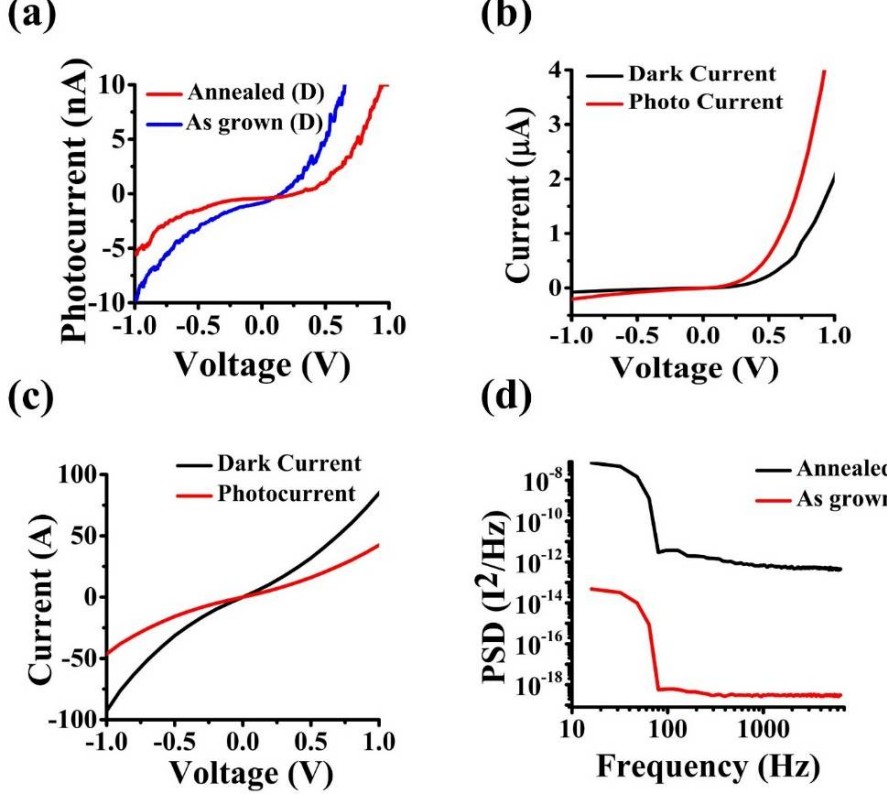

**Figure 4.** (**a**) SNW I-V characteristics of annealed and as-grown doped nanowires under 860 nm illumination. Ensemble NW IV characteristics of (**b**) as grown GaAs: Te NWs and (**c**) annealed GaAs: Te NWs, and (**d**) low-frequency noise (LFN) spectrum.

The temperature dependence of dark current was investigated to ascertain the activation energy of these traps. The temperature-dependent dark current variation of the as-grown and the annealed NWs at a forward bias of 0.2 V are depicted in Figure 5a,b, respectively. The activation energies in annealed and as grown Te-doped NWs were calculated from the slope of the Arrhenius plots as shown in Figure 5, using the following equation, assuming the presence of only one defect state:

$$I \sim e^{-\frac{E_A}{k_B T}} \tag{1}$$

where $I$ is the experimentally measured dark current, $E_A$ is the activation energy, $k_B$ is the Boltzmann constant, and $T$ is the absolute temperature. Equation (1) can be simplified as:

$$\ln(I) \sim -\frac{E_A}{k_B} \frac{1}{T} \tag{2}$$

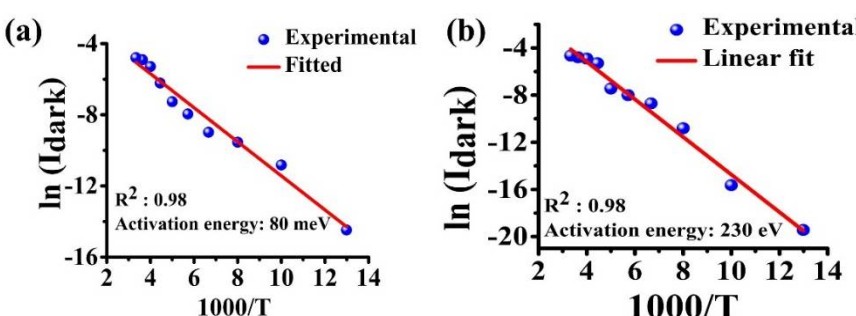

**Figure 5.** Temperature dependence of dark current at 0.2 V forward bias in (**a**) as-grown and (**b**) In-situ annealed Te-doped GaAs NWs.

Activation energies of 80 meV and 230 meV were determined from the best fit of the experimental I-V characteristics of the as-grown and annealed doped NWs, respectively. A larger activation energy value in the annealed NW is consistent with a deep-level defect in the bandgap that acts as an electron trap.

## 3. Experimental Details

### 3.1. Nanowire Growth

Te-doped GaAs NWs were grown using an EPI 930 solid-source MBE system by the self-catalyzed VLS growth mechanism. First, NWs were grown on a p-Si (111) substrate at 620 °C using a group Ga flux of $2.4 \times 10^{-6}$ Torr and a As flux of $4.8 \times 10^{-7}$ Torr. Growth was initiated by opening the Ga shutter 13 secs before opening the As shutter. Next, the Te-doping of the NWs was accomplished using a GaTe source at 550 °C, based on the optimization in our previous work [27]. The droplet termination was conducted by first closing the Ga shutter, then varying the As flux.

Three NW samples were grown for annealing investigations. Ga droplets were terminated under varying As flux. Subsequently, a passivating shell of AlGaAs was grown at the substrate temperature of 465 °C. The entire NWs with the shells were then In-situ annealed under an ultra-high-vacuum ambient (~$10^{-10}$ Torr) at 465 °C for 5 min.

### 3.2. Characterization Techniques

Morphological properties of the NWs were investigated using the Carl-Zeiss Auriga-BU FIB field emission scanning electron microscope (FESEM). A Thermo Fisher Talos F200X 80–200 kV scanning/transmission electron microscope (S/TEM), operating at 200 kV, was used for high-resolution imaging. The PL emission from the doped NWs was measured using a 633 nm He-Ne laser source, a double grating monochromator for wavelength dispersion, an InGaAs detector, and conventional lock-in amplifier techniques for PL emission detection. In addition, a closed-cycle optical cryostat from Montana Cryostation,

with the sample chamber interfaced with a fiber-coupled confocal microscope, was used to determine the PL characteristics at 4K. Conductive atomic force microscopy (C-AFM) in the contact mode was used for room temperature single nanowire I-V characteristics. The details of AFM measurement are reported elsewhere [27].

## 4. Conclusions

The As flux used for droplet consumption and post-growth annealing were found to strongly influence the optical and electrical properties of these self-catalyzed Te-doped GaAs NWs. An As flux exceeding the growth flux for termination resulted in stacking faults adversely impacting the optoelectronic NW properties. In-situ annealing of the doped NWs likewise negatively affected the electronic properties. This has been attributed to the annealing induced non-radiative recombination channels. Support for this hypothesis comes from the presence of a broad lower energy PL peak with a four-fold decrease in the overall PL intensity, along with negative photocurrent and significant enhancement of the noise level in low-frequency noise spectra of annealed and doped ensemble NWs in comparison to as-grown NWs. An activation energy of ~230 meV has been estimated for this non-radiative recombination center from temperature-dependent I-V measurements and is assigned to a $Te_{As}V_{Ga}^{-}$ complex acceptor. Thus, this study shows seed droplet consumption using the same As flux as the growth flux without any post-growth annealing is the optimized growth condition for attaining high-quality self-catalyzed Te-doped GaAs NWs.

**Author Contributions:** Design of experiments for nanowire growth, data analysis and writing an initial draft, S.D.; device fabrication for ensemble I-V measurement, M.P.; AFM measurement, P.R.; PL measurement, H.K.; HRTEM for microstructural properties study, A.P. and L.R.; Initial conceptualization of the project, funding acquisition, review of the first draft and editing, S.I. All authors have read and agreed to the published version of the manuscript.

**Funding:** This work is primarily based upon research supported by the Air Force Office of Scientific Research (AFOSR) under grant number W911NF1910002 with support and encouragement of Gernot Pomrenke. A part of this work was also supported by the National Science Foundation under grant# ECCS-1832117 (technical monitor: Dominique Dagenais).

**Data Availability Statement:** All the relevant data are included in this publisehd article.

**Acknowledgments:** This work was performed at the Joint School of Nanoscience and Nanoengineering, a member of the Southeastern Nanotechnology Infrastructure Corridor (SENIC) and National Nanotechnology Coordinated Infrastructure (NNCI), which is supported by the National Science Foundation (ECCS-1542174). TEM work was performed in part at the Analytical Instrumentation Facility (AIF) at North Carolina State University, which is supported by the State of North Carolina and the National Science Foundation (award number ECCS-2025064). The AIF is a member of the North Carolina Research Triangle Nanotechnology Network (RTNN), a site in the National Nanotechnology Coordinated Infrastructure (NNCI).

**Conflicts of Interest:** The authors declare no conflict of interest.

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
