# Peer review of "A Study on the Effects of Gallium Droplet Consumption and Post Growth Annealing on Te-Doped GaAs Nanowire Properties Grown by Self-Catalyzed Molecular Beam Epitaxy"

_catalysts, doi:10.3390/catal12050451_

Round 1

Reviewer 1 Report

The present manuscript reports the preparation and characterization of Te-doped GaAs nanowires grown by a self-catalyzed vapor liquid-solid (VLS) growth mechanism using molecular beam epitaxy. The authors investigated systematically the parameters that influence the growth of the relevant nanowires and their manufacturing quality that consequently affect their optical and electrical properties. More specifically the authors identified that the amount of As flux used for consumption before the termination of the NW core growth has a direct impact on the density of stacking faults at the NW tip. Interestingly, post=growth annealing affects the NWs were characterized by field emission scanning electron microscope (FESEM), scanning/transmission electron microscope (S/TEM),  PL emission at room and temperature, and 4K and conductive atomic force microscopy (C-AFM). The manuscript is well written and referenced.

Just a minor point for the authors to consider.

  1. Page line 116: Please define FWHM and SNW.

The present manuscript demonstrates experimentally the structural effects of experimental parameters on the NW growth which consequently impacts the optoelectronic performance of the manufactured NW while the authors make an effort to shed light on the reasons why this might be happening. I do believe that these details are important in relation to the manufacturing of useful material with applications in electronics and will attract the interest of material chemists and engineers. I am happy to recommend the publication of the manuscript after minor revision.

Reviewer 2 Report

The paper presented by Devkota and co-workers is very interesting and well presented and it requires only minor modifications before its publication according to the following comments:

1) In the abstract too many abbreviations are present and this makes difficult the reading. Please, specify what As, Ga, PL and Te mean.

2) Check the reference. I think that it is required to insert the whole references in the square brackets.

3) The authors should improve the section of Experimental details adding a paragraph for the employed materials and separating in two sections the procedure of NWs preparation from the procedure for the properties investigation.

4) Which are the three As fluxes adopted for droplet? They are not specified.

5) In Figure 5 the authors should report in the two graphs the R2 of the fitting and the values of the calculated activation energies.
